# Cross-Cultural Analysis of Human Values, Morals, and Biases in Folk Tales

**Winston Wu[1,2]**    **Lu Wang[2]**    **Rada Mihalcea[2]**

[1]Department of Computer Science, University of Hawai'i at Hilo
[2]Computer Science and Engineering, University of Michigan – Ann Arbor
wswu@hawaii.edu, {wangluxy,mihalcea}@umich.edu

## Abstract

Folk tales are strong cultural and social influences in children's lives, and they are known to teach morals and values. However, existing studies on folk tales are largely limited to European tales. In our study, we compile a large corpus of over 1,900 tales originating from 27 diverse cultures across six continents. Using a range of lexicons and correlation analyses, we examine how human values, morals, and gender biases are expressed in folk tales across cultures. We discover differences between cultures in prevalent values and morals, as well as cross-cultural trends in problematic gender biases. Furthermore, we find trends of reduced value expression when examining public-domain fiction stories, extrinsically validate our analyses against the multicultural Schwartz Survey of Cultural Values, and find traditional gender biases associated with values, morals, and agency. This large-scale cross-cultural study of folk tales paves the way for future studies on how literature influences and reflects cultural norms.

## 1 Introduction

Folk tales are stories told in the oral tradition. All cultures tell stories, though the manner in which the stories are told can differ (McCabe, 1997). According to folklorist William Bascom (1954), folktales have four purposes: (1) to let people escape from society, (2) to validate and justify culture, (3) to reinforce morals and values, and (4) to apply social pressure. Most of the previous work on folk and fairy tales has focused on a limited set of Anglo-American or European cultures, ignoring large swaths of the world's diverse cultures. In this paper, we present a *cross-cultural* analysis of folk tales covering a wide range of cultures around the world, focusing on Balcom's latter two purposes.

Specifically, we examine the values, morals, and gender biases expressed in folk tales, quantifying similarities and differences in how these aspects are expressed across cultures. To do so, we first compile a large corpus of over 1,900 folk tales originating from 27 cultures from multiple online sources. We then employ lexicon-based correlation analyses to investigate human values in folk tales across cultures. To extrinsically validate our analyses, we compare values expressed in folk tales with values expressed in long-form novels, as well as cultural values from the Schwartz Cultural Values Orientation Survey (Schwartz, 2006). In a similar manner, we analyze morals in folk tales, using the framework of Moral Foundation Theory (Graham et al., 2013, 2018). Further, we investigate the representation of male and female characters in folk tales, as well as gender associations with values, morals, and verbs of agency and power. Our analyses of values, morals, and gender bias in folk tales across cultures reveal statistically significant differences between cultures, as well as cross-cultural trends, which we detail in the following sections.

This paper presents several novel contributions: (1) the collection of a large dataset of folk and fairy tales from across 27 cultures, (2) the application of automatic lexicon-based measurements of values and morals to folk tales across cultures, which have traditionally been manual efforts, (3) cross-cultural analyses of these folk tales with respect to values and morals, revealing values corresponding to human connection, and (4) cross-cultural analyses of gender biases with respect to values, morals, agency, and power, revealing associations with traditional gender roles. To our knowledge, this is the largest cross-cultural computational study of folk tales, covering cultures from six continents. Code and data for reproducing our experiments can be found at github.com/wswu/folktales.

## 2 Data

Folk tales have largely been passed down through oral tradition but have recently been digitized. We

compile a large dataset of folk tales across multiple cultures from three online sources: the Alishman Folk Texts (AFT; Hagedorn and Darányi, 2022), the Multilingual Folk Tale Database[1] (MFTD), and Project Gutenberg.[2] AFT is a digitized compilation of tales collected and annotated by the late Professor D. L. Alishman. The Multilingual Folk Tale Database is a collection of tales shared by various users in multiple languages. While these are the largest available collections of folk tales online, they are largely limited to Euro-centric cultures. To enable a wider analysis across cultures and cover all six continents of the world (except Antarctica), we augment this set with tales from public domain tale compilations from Project Gutenberg, representing the following cultures: Africa, Australia, Brazil, Canada, China, Japan, and the Himalayas. The list of sources for these tales can be found in Table 1. We manually cleaned each text by removing Project Gutenberg boilerplate text, image captions, and excess whitespace.

We combine tales from these three data sources with the following modifications: (1) we consider tales whose provenance in AFT is "Aesop" to be from Greece;[3] (2) we remove duplicate tales (the culture of origin is identical and the Jaccard similarity between the lowercased words in the title is greater than 0.8); (3) we remove tales shorter than 50 or longer than 4,000 tokens (for speed; 13 tales were removed); (4) for adequate representation, we only keep tales when there are at least 10 tales from that culture; (5) to facilitate our analysis, we remove tales not available in English. We emphasize that while the text is in English, the content of the tales is representative of the tales' culture. For tales not originally written in English, some tales were compiled in English based on interactions with native speakers, while others were translated into English from existing sources. To our knowledge, all folk tale compilers and translators were professional folklorists who have dedicated their lives to studying a specific culture, and thus have an intimate understanding of the culture of the tale, as well as native-level proficiency in English and the original language.

In total, our dataset comprises 1,925 tales across 27 cultures. 788 tales come from AFT, 880

---

[1]www.mftd.org
[2]www.gutenberg.org
[3]Folk tales do not include mythology.

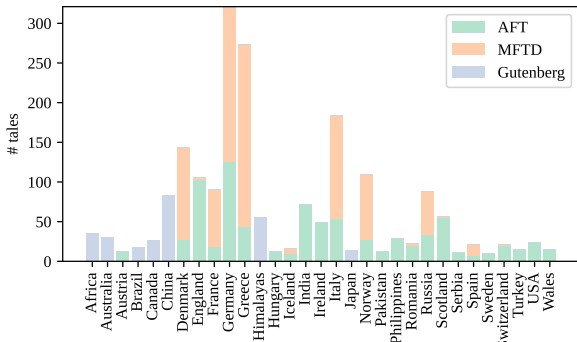

Figure 1: Number of fairy tales per culture.

tales come from MFTD, and 257 tales come from Project Gutenberg, with an average of 1,036 words per tale. Figure 1 presents the total number of tales from each culture.

## 3 Human Values

One of the functions of folklore is to teach and reinforce values (Bascom, 1954). Values are inextricably connected with culture, and naturally, people in different cultures hold different values. A long line of research by social psychologist Shalom Schwartz has proposed the Theory of Basic Human Values (Schwartz and Bilsky, 1987; Schwartz, 2012), which are universal guiding principles for a person or group. Schwartz originally identified 10 human values (Schwartz and Bilsky, 1987), later 19 (Schwartz et al., 2012), which were validated to be comprehensive across cultures through surveys in over 65 countries. Values and morals in fairy tales have also been manually analyzed and annotated, especially in the education literature (e.g. Moss, 1988; Guroian, 1996). However, these studies are usually limited to a small set of tales of interest. In addition, existing work measuring cross-cultural values (Rokeach, 1973; Inglehart et al., 2000) often rely on surveys, which are time-consuming and require human labor.

To enable the analysis of folk tales at a large scale, we propose an automated lexicon-based method to measure human values. We use the Values Lexicon, a crowd-sourced lexicon of 49 human values (Wilson et al., 2018) containing 1,267 English words and phrases. The Values Lexicon was constructed based on crowdsourcing by users from the United States, India, Kenya, the Philippines, and Trinidad and Tobago. It has been validated against human-labeled word intrusion and category matching tasks with high accuracy. The

| Origin | Title | Author/Editor |
|--------|-------|---------------|
| Japan | Japanese Fairy Tales | Yei Theodora Ozaki |
| Himalayas | Simla Village Tales, or, Folk Tales from the Himalayas | Alice Elizabeth Dracott |
| China | A Chinese Wonder Book | Norman Hinsdale Pitman |
| China | The Chinese Fairy Book | Richard Wilhelm |
| Africa | West African Folk-Tales | W. H. Barker and Cecilia Sinclair |
| Australia | Australian Fairy Tales | Atha Westbury |
| Canada | Canadian Fairy Tales | Cyrus Macmillan |
| Brazil | Fairy Tales from Brazil: How and Why Tales from Brazilian Folk-Lore | Elsie Spicer Eells |

Table 1: Folk tales compiled and cleaned from Project Gutenberg.

| | | |
|---|---|---|
| Accepting-others | Achievement | Advice |
| Animals | Art | Autonomy |
| Career | Children | Cognition |
| Creativity | Dedication | Emotion |
| Family | Feeling-good | Forgiving |
| Friends | Future | Gratitude |
| Hard-work | Health | Helping-others |
| Honesty | Inner-peace | Justice |
| Learning | Life | Marriage |
| Moral | Nature | Optimism |
| Order | Parents | Perseverance |
| Purpose | Relationships | Religion |
| Respect | Responsible | Security |
| Self-confidence | Siblings | Significant-other |
| Social | Society | Spirituality |
| Thinking | Truth | Wealth |
| Work-ethic | | |

Table 2: The 49 values from the Values Lexicon (Wilson et al., 2018).

list of values in the Values Lexicon is shown in Table 2. We use this lexicon to analyze the human values expressed in folk tales.

We analyze folk tales for human values using the following process. First, we tokenize and lowercase each tale with spaCy.[4] We then compute a distribution of human values for each tale by counting tokens associated with a human value and normalizing by the total number of tokens in the text. We compare and analyze these distributions across cultures to answer several research questions below.

**What values are expressed in folk tales?** To identify the most common values expressed in folk tales, we examine the top three most prevalent values within each tale. Across all cultures, the most common values are Social values, followed by Society and Relationships, as shown in Figure 2, computed by examining the top three most prevalent values within each tale. The Social value contains words like mother, father, son, together, house, home, family, and live. Many words in the Social value overlap with Society and Relationships words. These values reveal that across cultures, *folk tales emphasize values related to human connection*. Several other prevalent values, including Relationships, Helping-others, Family, Significant-other, and Children, are directly related to interpersonal relationships. This is significant because humans are wired to form bonds with one another (Fishbane, 2007). Many folk tales, which were originally orally passed down, may have been created to reinforce the importance, or even usefulness, of human connections in past societies.

**What are the most common values by culture?** Next, we examine whether there are differences in the values expressed in folk tales from different cultures. Figure 2 plots the top 10 values expressed in each culture's tales, where the dark color indicates that the value is more prevalent (the full figure with all values is presented in Figure 6). A visual inspection reveals that there are differences within cultures, and this is natural because each culture has unique societal norms (Schwartz, 1999). For example, research has shown that there are differences in basic societal values between Europe and Asia (Blondel and Inoguchi, 2006; Hofstede, 2007). Indeed, our analysis shows that the cultures of the Himalayas, Hungary, and the Philippines show the strongest expression of Social values, compared to Denmark, France, and Greece, which show the weakest expression.

To test whether these differences are statistically significant, we perform multiple two-sided T-tests, with Bonferroni correction, because we are making thousands of comparisons. We find a total of 229 statistically significant differences ($p \leq 0.05$) when comparing any two culture-value pairs. For example, Hungary's expression of Social values is statistically significantly different from Italy, Norway, Denmark, France, and Greece. While these

---

[4] spacy.io

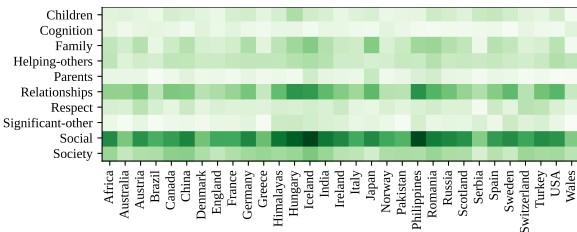

Figure 2: The top 10 human values in fairy tales across cultures. The darker the color, the more prevalent that value is expressed in that culture.

results are significant, the interpretation of these results requires in-depth knowledge of the respective cultures. For example, the Philippines is said to have "no culture" but had a strong social hierarchy before being colonized by Spain (Rosaldo, 1988; Junker, 1990), which may explain the high expression of Social words in its tales. We emphasize that these kinds of large cross-cultural analyses of values are possible without the use of surveys, as is the norm in the existing literature on cross-cultural values, but will require collaboration with cultural anthropologists and other social scientists to fully understand the findings from such analyses.

**How do folk tales compare to fiction books?**
One outstanding question is whether the values expressed in folk tales are also reflected in the broader literature. Tales are often written to teach a lesson or explain how the world works, but they usually depict fictional events. Fiction novels are also fictional, but the purpose of fiction is often to entertain (Follett, 1923). Perhaps the authors who grew up in a culture perpetuate values in their own works that they learned from that culture's folk tales. To investigate this question, we compile a set of public-domain fiction books written in English from Project Gutenberg whose authors were born in a country covered by our folk tale dataset. This set of fiction comprises 260 books across 9 cultures: Australia, England, France, Germany, Ireland, Italy, Russia, Spain, and USA. We then perform the same analysis of values on this dataset of novels.

As these books are substantially longer than folk tales (on average, 92,845 tokens per novel vs. 1,036 tokens for folk tales), we first examine whether there is a difference in the percentage of words expressing values in folk tales and longer-form novels. Table 3 presents the percentage of value-expressing words in folk tales and novels

| Culture | Tales | Gutenberg |
|---|---|---|
| Africa | 5.06 | |
| Australia | 4.52 | 4.11 |
| Austria | 5.09 | |
| Brazil | 3.60 | |
| Canada | 4.86 | |
| China | 5.22 | |
| Denmark | 4.35 | |
| England | 4.32 | 3.96 |
| France | 5.05 | 3.84 |
| Germany | 4.84 | 3.49 |
| Greece | 4.39 | |
| Himalayas | 5.28 | |
| Hungary | 5.52 | |
| Iceland | 5.84 | |
| India | 5.23 | |
| Ireland | 4.66 | 4.05 |
| Italy | 4.45 | 3.69 |
| Japan | 4.46 | |
| Norway | 3.77 | |
| Pakistan | 3.62 | |
| Philippines | 5.05 | |
| Romania | 5.32 | |
| Russia | 4.91 | 5.91 |
| Scotland | 4.38 | |
| Serbia | 3.42 | |
| Spain | 4.75 | 4.92 |
| Sweden | 4.63 | |
| Switzerland | 4.64 | |
| Turkey | 4.86 | |
| USA | 4.47 | 4.03 |
| Wales | 4.09 | |
| Mean | 4.64 | 3.99 |

Table 3: Percentage of words in folk tales and Gutenberg fiction novels that express values. Overall, novels contain a smaller percentage of value words compared to folk tales.

from Project Gutenberg. On average, 4.64% of words in folk tales are value-expressing, compared to 3.99% in novels. Thus, *across cultures, novels express values at a lower rate than folk tales.* This supports Bascom's (1954) notion that one purpose of folk tales is to reinforce values, in comparison to novels, which are primarily written to entertain a broader target audience.

Next, we examine if there is a *change* in values expressed between folk tales and longer novels. To do this analysis, we only consider value-expressing words, comparing the average expression of values between folk tales and longer novels. We find that even after controlling for the length of the novels, the expression of most values decreases. Figure 7 presents the top five expressed values. A marked decrease in the percentage of value-expressing words across different values reinforces that novels express values at a lower rate compared to folk tales.

**Are values in folk tales reflective of broader culture?** We also seek to extrinsically validate the values in folk tales. In addition to human values, Schwartz also studied cultural values (Schwartz, 2006). This cross-cultural study identified seven cultural values — Hierarchy, Embeddedness, Mastery, Affective Autonomy, Intellectual Autonomy, and Egalitarianism — by surveying K-12 teachers and college students in 80 countries, who provided scores for these cultural values ranging from 1 and 5. For our analysis, we identify human values that most closely align with these Cultural Values Orientation Scores from Schwartz's cultural value survey. We examine 19 countries[5] which are covered by Schwartz's study and also represented in our folk tales. For each pair of cultural value and human value, we rank the countries by their cultural value score as well as by our average human value measurement in their folk tales. Next, we compute the Spearman rank correlation between these two rankings. The most positively and negatively correlated cultural and human values are presented in Table 4.

We briefly interpret some of these findings. Affective Autonomy refers to the pursuit of "affective positive experiences", such as enjoyment of life (Schwartz, 2006). China, India, and Turkey are the countries with the highest Affective Autonomy score. The Wealth value is strongly correlated with Affective Autonomy, because wealth gives one the means to enjoy life. In contrast, the Children value is negatively correlated; perhaps having children takes away time for these positive affective pursuits. The Hierarchy and Marriage values also have a natural explanation. In hierarchical societies, individuals are socialized to conform to the roles of the hierarchy (Schwartz, 2006). The act of marriage comes with roles and responsibilities that must be upheld. Unexpectedly, the Respect value is strongly negatively correlated with Hierarchy. Since the Respect value is associated with words related to respecting (respect, honoring) and the older generation (elders, seniority), we would expect that respecting authority is an integral part of a hierarchical society. This may indicate that human values and cultural values are not always aligned.

| Cultural Value | Human Value | $\rho$ |
|---|---|---|
| AffAuton | Wealth | 0.65 |
| Hierarchy | Marriage | 0.52 |
| Mastery | Nature | 0.50 |
| Mastery | Self-confidence | 0.43 |
| Harmony | Family | 0.41 |
| AffAuton | Children | -0.56 |
| Embedded | Career | -0.60 |
| Egalitarianism | Work-ethic | -0.62 |
| Embedded | Siblings | -0.64 |
| Hierarchy | Respect | -0.77 |

Table 4: The most positively and negatively correlated cultural values and human values computed by ranking 19 countries. AffAuton refers to Affective Autonomy. $\rho$ is Spearman's rho.

## 4 Moral Foundations

Folk tales also teach morals (Bascom, 1954), and culture is an important variable that shapes one's morals (Prinz, 2014). To examine morals in folk tales, we employ Moral Foundations Theory (MFT; Graham et al., 2013, 2018), which was created by social psychologists to explain the basis of moral actions. MFT defines five moral foundations,[6] each with a positive and negative aspect: Care/Harm, Fairness/Cheating, Loyalty/Betrayal, Authority/Subversion, and Sanctity/Degradation. MFT has recently been used to analyze a variety of text, most notably in social media and politics (e.g. Fulgoni et al., 2016; Roy and Goldwasser, 2021; Asprino et al., 2022). Analogous to our analysis of human values, we perform a lexicon-based analysis of moral foundations on folk tales by employing the Moral Foundations Dictionary 2.0 (Frimer et al., 2019), which associates 2,103 words and phrases with the 10 moral foundations in MFT. Figure 3 displays the distribution of cultures expressing moral foundations through folk tales.

**Are there cross-cultural trends in moral foundations expressed in folk tales?** We first examine cross-cultural trends. Overall, we find a stronger expression of positive dimensions (Care, Loyalty, Authority, and Sanctity) compared to the negative dimensions. This may be because positive aspects of morality ("thou shalt") promote human welfare in comparison to negative aspects

---

[5]Austria, China, Denmark, France, Greece, Hungary, India, Ireland, Italy, Japan, Norway, Pakistan, Philippines, Romania, Russia, Serbia, Spain, Sweden, and Turkey.

[6]A later version of this theory defines a sixth dimension, Liberty/Oppression, which we do not consider because available lexicons do not cover this dimension.

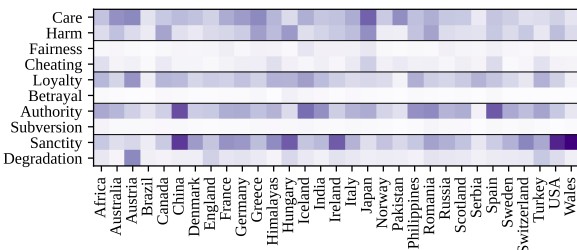

Figure 3: Moral foundations expressed in fairy tales across cultures.

| China | | Spain | |
|---|---|---|---|
| Word | # Tales (%) | Word | # Tales (%) |
| order | 41 (0.08) | ordered | 9 (0.14) |
| father | 34 (0.07) | father | 7 (0.11) |
| master | 29 (0.06) | order | 6 (0.09) |
| emperor | 23 (0.05) | master | 5 (0.08) |
| ordered | 23 (0.05) | commanded | 4 (0.06) |
| bowed | 23 (0.05) | servants | 4 (0.06) |
| servants | 18 (0.04) | servant | 4 (0.06) |
| honor | 14 (0.03) | elder | 3 (0.05) |
| queen | 13 (0.03) | respected | 2 (0.03) |
| willing | 13 (0.03) | protect | 2 (0.03) |

Table 5: The most frequent words associated with the Authority moral dimension from Chinese and Spanish folk tales, and the number of tales including each word. Though historically China and Spain were both monarchies, the emperor is mentioned in a larger proportion of Chinese folk tales.

("thou shalt not") (Staub, 2013). As tools for teaching, folk tales need to convey their message, and positive interactions are a more effective way to do so. In addition, we find that the Fairness/Cheating dimension is more weakly expressed compared to the other dimensions.

Looking across cultures, we identify statistically significant differences in the expressions of morality by performing multiple T-tests with Bonferroni correction. Our analysis identified 54 statistically significant differences. The full listing of these culture pairs is presented in Figure 8. As a case study, we consider folk tales from China, which shows significant differences from other cultures in its expression of Authority. It is well known that Chinese culture values deference to authority (Hwang, 2012; Zhai, 2017). How is this value reflected in folk tales? We examine the most frequent words associated with the Authority dimension in Chinese and Spanish folk tales, shown in Table 5, because tales from both of these cultures express strong Authority. We find that *order*, both as a noun and a verb, *father*, and *master* are the top Authority terms in both Chinese and Spanish folk tales. However, the next most frequent term, *emperor*, occurs in over 5% of Chinese folk tales, but does not appear in the top terms for Spanish tales. Historically, both China and Spain were monarchies. However, in Chinese tradition, the emperor is traditionally believed to be of divine descent (Werner, 2020). Because of this, the emperor features prominently in tales, similar to other deities in tales from other cultures. Spanish tales, on the other hand, often feature a king. Thus, an analysis of morals from folk tales can also reveal historical phenomena in culture.

## 5 Gender Bias

Finally, we turn our focus to gender bias, which is pervasive in children's literature (Louie, 2001). Studies have shown that gender stereotypes are picked up by the children reading the tales. For example, in a study of primary-school children who read Cinderella, girls did not identify with Cinderella due to her lack of independence, but boys identified with Prince Charming due to his independence (Westland, 1993). An investigation of gender stereotypes in 247 children's books indicated that gender bias is indicative of the target audience (Lewis et al., 2022). Other studies show that in children's books, mothers play a dominant role in parenting, while fathers are underrepresented (Anderson et al., 2021). Some of these gender discrepancies disappear if the author is female (Nagaraj and Kejriwal, 2022). There has been growing interest in investigating gender bias in children's literature, specifically children's books, but there is little work on large collections of folk tales across multiple cultures. Our work seeks to rectify this shortcoming in the literature.

To support the analyses in this paper, we first identify the gender of characters in each tale. We follow the process in Lucy and Bamman (2021) of performing dependency parsing on each tale using spaCy with the en_core_web_sm model and coreference resolution using fast-coref (Toshniwal et al., 2021). Within each coreference cluster, we assign the character's gen-

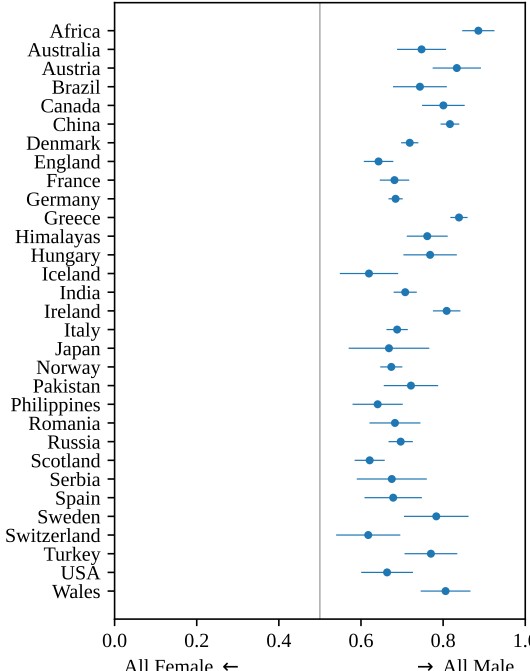

Figure 4: Average gender ratios of characters in folk tales across cultures. An average ratio of 0.72 across all cultures indicates that a tale has on average 2.57 male characters for every female character. Error bars indicate standard error.

der based on which gendered pronoun is more frequent: male (he/him/his/himself) or female (she/her/hers/herself). We only consider characters if they have at least three mentions in a story. We now investigate the representation of male and female characters in the story.

**How are genders represented in folk tales?** Across cultures, *stories simply have more male characters than female characters.*[7] In total, our dataset contains 1,680 female characters and 3,853 characters (a 2.29x difference). When computing gender proportions on a per-tale basis, we find that a tale has on average 2.57 male characters for every female character. Figure 4 shows the stark differences in the ratio of male characters to female characters in folk tales. Similar gender disparities have been found in fairy tales (Zhou et al., 2022), children's picture books (Gooden and Gooden, 2001), and movies (Smith et al., 2015).

**What is the relationship between gender and values?** While some folk tales were written for educational value, Zipes (2012) argues that many

---

[7]Many tales have animals as characters, but these animals are anthropomorphized and given a gender.

tales were written to teach boys and girls different values and attitudes appropriate for their gender. To determine which differences, we examine value-expressing words that are associated with mentions of gendered characters. Specifically, we compute the co-occurrence of value-expressing words within a 5-word window of each mention of gendered characters. Table 6 presents the percentage of male and female characters associated with each value. We find that on average, *female characters are associated with values that are traditionally considered feminine*, such as Children, Family, and Marriage (Fox and Murry, 2000). In contrast, male characters have significant differences in traditionally masculine values such as Career, Wealth, Friends, and Helping-others, and cognitive values such as Cognition, and Thinking.

Table 6 contains a listing of values associated with female and male characters in folk tales.

**What is the relationship between gender and moral foundations?** Using the same procedure for analyzing gender and human values, we compute the association between gender and moral foundations (Table 7). We find that in folk tales, female characters are strongly associated with the Care dimension, while male characters are more associated with Authority. This seems to corroborate existing findings that women are more concerned about preventing harm (Efferson et al., 2017) and that women are depicted as caring and nurturing, while men are depicted as authority figures (Tronto, 1995). We also find that *male characters are more associated with the dimensions of Cheating, Betrayal, and Degradation*; the differences are smaller but still statistically significant. These may correspond with the "bad guy" trope that occurs frequently in tales (Davies, 1995).

**What is the relationship between gender, agency, and power?** Characters can be perceived to have agency and power based on the actions they take. As an example from Sap et al. (2017), consider the verbs "accept" and "assess". A character who accepts things is perceived to be more passive (lower agency) than someone who assesses. Sap et al. (2017) found that male characters are portrayed with more agency than female characters in movie scripts. Can we also see these differences in folk tales? Following Sap et al. (2017), we perform dependency parsing on our folk tales using spaCy to identify verbs, the sub-

| Value | Female % | Male % |
|---|---|---|
| Accepting-others | 0.49 | 0.46 |
| Achievement | 0.11 | **0.32** |
| Advice | 0.62 | 0.76 |
| Animals | 0.99 | **1.86** |
| Art | 0.65 | 0.79 |
| Autonomy | 0.07 | 0.08 |
| Career | 0.95 | **1.48** |
| Children | **13.41** | 9.84 |
| Cognition | 5.31 | **6.32** |
| Creativity | 0.06 | 0.08 |
| Dedication | 0.06 | 0.08 |
| Emotion | 0.01 | 0.04 |
| Family | **24.01** | 14.81 |
| Feeling-good | 2.58 | 2.73 |
| Forgiving | 0.20 | 0.22 |
| Friends | 0.86 | **1.68** |
| Future | 0.81 | 0.89 |
| Gratitude | 1.16 | 1.34 |
| Hard-work | 0.35 | 0.36 |
| Health | 0.10 | 0.11 |
| Helping-others | 9.02 | **10.96** |
| Honesty | 0.07 | 0.14 |
| Inner-peace | 0.80 | 0.75 |
| Justice | **0.94** | 0.67 |
| Learning | 0.66 | **1.14** |
| Life | 3.74 | 4.02 |
| Marriage | **12.62** | 6.34 |
| Moral | 0.06 | 0.14 |
| Nature | 0.08 | 0.17 |
| Optimism | 0.20 | 0.24 |
| Order | 1.41 | 1.47 |
| Parents | **10.36** | 5.90 |
| Perseverance | 0.21 | 0.26 |
| Purpose | 0.42 | 0.69 |
| Relationships | **37.19** | 25.05 |
| Religion | 2.39 | **3.57** |
| Respect | 6.64 | **8.62** |
| Responsible | 0.10 | 0.13 |
| Security | 0.33 | 0.52 |
| Self-confidence | 0.11 | 0.11 |
| Siblings | **4.07** | 2.57 |
| Significant-other | **14.05** | 7.53 |
| Social | **47.86** | 36.38 |
| Society | 11.93 | **15.06** |
| Spirituality | 1.39 | 1.47 |
| Thinking | 4.77 | **5.58** |
| Truth | 1.03 | **1.68** |
| Wealth | 1.85 | **3.39** |
| Work-ethic | 0.75 | 0.92 |

Table 6: Percentage of female and male characters associated with human values. A number is bolded if it is greater than the opposite gender, and this difference is statistically significant. Female characters on average are more associated with values that are traditionally feminine.

| Foundation | Female % | Male % |
|---|---|---|
| Care | **26.91** | 14.61 |
| Harm | 10.45 | 11.34 |
| Fairness | 2.30 | 2.30 |
| Cheating | 2.51 | **3.82** |
| Loyalty | 15.15 | 14.55 |
| Betrayal | 0.35 | **0.90** |
| Authority | 15.57 | **21.73** |
| Subversion | 1.00 | 0.99 |
| Sanctity | 14.04 | 14.91 |
| Degradation | 3.36 | **4.60** |

Table 7: Percentage of female and male characters associated with moral foundations. A number is bolded if it is greater than the opposite gender, and this difference is statistically significant.

ject of the verb (dependency relation *nsubj*), and object of the verb (dependency relation *nsubjpass* and *dobj*). For this analysis, we use a lexicon of 2,155 verbs annotated for agency (positive, negative, equal) and power (agent, theme, equal) (Sap et al., 2017).

We first examine agency. After accounting for the difference in number of male/female characters, we find that *male characters are more likely than female characters to be agents of a verb* Figure 5. The verbs themselves also warrant investigation; for some verbs, the agent plays a more active role, whereas for other verbs, the agent plays a more passive role. To investigate the interplay between gender and agency, we count the gender of verbs' agents. The results are presented in Table 8. We find that overall, positive agency verbs are more prevalent than negative agency words. When separated by the gender of the agent, male characters are more likely to be agents of a positive agency verb, while female characters are more likely to be agents of a negative agency verb. Our findings across cultures are similar to Sap et al. (2017)'s findings on movie scripts.

Next, we examine power. Sap et al. (2017) annotates verbs for power in one of three ways: either the agent has stronger power than the theme, the theme (object) has stronger power than the agent, or the power between the agent and theme are equal. We examine the association between male and female characters as agents and themes of agent-powerful vs. theme-powerful verbs in Table 9. For Agent-powerful verbs, we find that male characters are 2.67x more likely to be the agent than the theme (compared to 2.10x for female char-

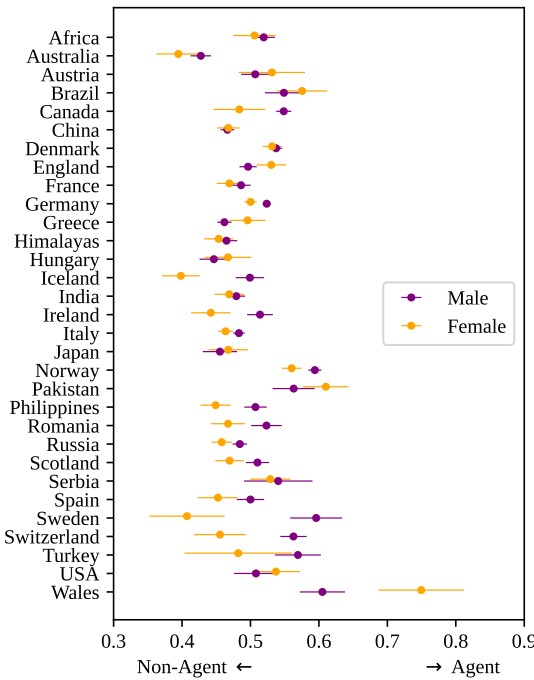

Figure 5: Proportion of male/female characters as agents of events. Across cultures, male characters are more likely to be the agent.

| Agency | Female | Male |
|---|---|---|
| Positive | 16,687 (0.65) | 39,851 (**0.67**) |
| Negative | 6,108 (**0.24**) | 13,151 (0.22) |
| Equal | 3,037 (0.12) | 6,846 (0.11) |

Table 8: Association between agency and gender. *Positive* means that the agent has an active role in the event, while *Negative* means the agent plays a more passive role. Male characters are more likely to be agents of a positive agency verb, while female characters are more likely to be agents of a negative agency verb.

acters. For Theme-powerful verbs, male characters are 2.46x more likely to be the agent than the theme of an Agent-powerful verb (compared to 2.32x for female characters). This decrease in the ratio between male and female characters indicates a gender imbalance: male characters are more likely to be agents of an agent-powerful verb, than female characters. Again, this corroborates (Sap et al., 2017)'s findings.

# 6 Conclusion

This paper presented a large-scale cross-cultural analysis of values, morals, and gender bias in folk tales. We first collected a large dataset of over 1,900 folk tales from 27 diverse cultures around the world. Building upon well-known theories of

|  | A>T | T>A | Equal |
|---|---|---|---|
| F/Agent | 9985 (0.21) | 3504 (0.21) | 9750 (0.24) |
| F/Theme | 4760 (0.10) | 1504 (0.09) | 2849 (0.07) |
| M/Agent | 24075 (0.50) | 8158 (0.49) | 22613 (0.56) |
| M/Theme | 9016 (0.19) | 3317 (0.20) | 4963 (0.12) |

Table 9: Association between power and gender. *A>T* means the verb is one where the agent has more power than the theme in the event, while *T>A* means the theme has more power than the agent. Male characters are more likely than female characters to be agents of Agent-powerful verbs.

Basic Human Values and Moral Foundations, we analyzed word usage in folk tales to identify cross-cultural trends of values involving human connection, as well as culture-specific preferences in the values and morals expressed in folk tales. Our analysis of gender bias in folk tales reveals that across cultures, male and female characters in the tales are associated with values and morals that are traditionally thought of as masculine and feminine, respectively. In addition, we found male characters have more agency in events than female characters.

This large-scale cross-cultural study of folk tales paves the way forward for future studies on how literature influences and reflects cultural norms. While folk tales are a reflection of their respective cultures, they can also have profound influences hundreds of years down the road on children reading these tales. Understanding the true effect of reading folk tales may require unrealistic longitudinal studies. However, machines are making great strides in language understanding. One interesting avenue of future research is to use NLP models to create new folk tales. Indeed, there have been some (manual) efforts to purposefully create or rewrite stories that subvert established gender biases (Crew, 2002). With recent advances in language modeling, we would like to automatically write stories using controllable generation and style transfer techniques. The automatic production of such debiased literature will hopefully bring awareness to long-standing biases that pervade our society.

# Acknowledgments

We thank the anonymous reviewers for their constructive feedback, and the members of the Language and Information Technologies (LIT) and LAnguage Understanding and generatioN re-

searCH (LAUNCH) labs at the University of Michigan for the insightful discussions during the early stage of the project. This project was partially funded by an award from the Templeton Foundation (#62256), a grant from the Air Force Office of Scientific Research (#FA9550-22-1-0099), and a grant from the Department of State (#STC10023GR0014). Any opinions, findings, and conclusions or recommendations expressed in this material are those of the authors and do not necessarily reflect the views of the Templeton Foundation or the Air Force Office of Scientific Research or the Department of State.

## Limitations

A selection of folk tales cannot fully represent the morals, values, and biases of a culture. Unfortunately, the analyses in this paper are also limited by the availability of folk tales online, which do not span every culture in the world. We also recognize that the coreference-based heuristic for gender inference has limitations (Cao and Daumé III, 2020), though it is an established approach for inferring characters' genders. In addition, this study analyzes folk tales written in English, due to the lack of existing tools in other languages for analyzing values and morals. As mentioned in Section 2, the texts of these tales, though written in English, are indeed representative of their original culture. For tales not originally written in English, some tales were compiled in English based on interactions with native speakers, and others were translated into English from existing sources. To our knowledge, all compilers and translators of the tales in our dataset were professional folklorists who have dedicated their lives to studying a specific culture, and thus have an intimate understanding of the culture of the tale, as well as native-level proficiency in English and the original language. Future work can investigate translating existing into English, or developing new lexicons or more sophisticated tools for analyzing tales in other languages.

## Ethics Statement

Folk tales have an important connection with culture. This work is making important progress in understanding how folk tales reflect culture as well as how folk tales can influence writers. Understanding the interplay between tales and culture will allow us to better understand the biases and values in our culture, and develop more socially just and equitable language technologies.

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

# A    Appendix

The appendix contains figures that were not able to fit in the main content. Figure 6 shows a heatmap of values expressed across all cultures. Figure 7 presents the top five expressed values in novels and folk tales. Figure 8 shows statistically significant differences in Moral word expression between cultures.

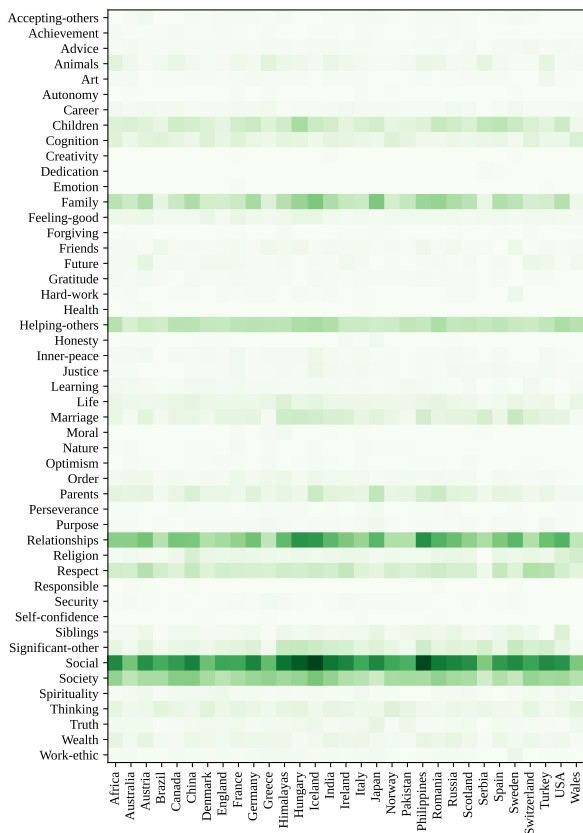

Figure 6: Complete listing of human values in fairy tales across cultures. The darker the color, the more prevalent that value is expressed in that culture.

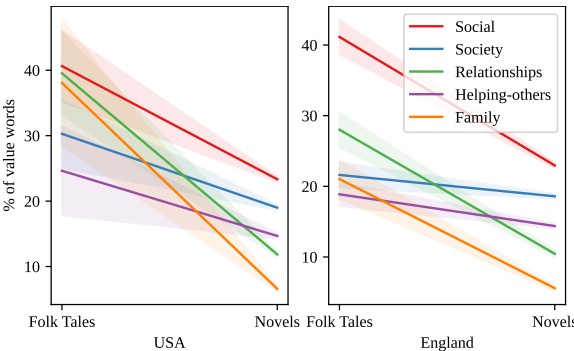

Figure 7: Percentage of value words representing the top 5 values. A value of e.g. 40 indicates that 40% of value-expressing words express that particular value. Note that these values may add up to more than 100% because a word can be associated with more than a single value. Error bars denote standard error.

| Moral | Culture1 | Culture2 | t-statistic | p-value |
|---|---|---|---|---|
| Harm | China | Hungary | -4.23 | 0.026 |
| Harm | China | Canada | -5.24 | <0.001 |
| Harm | Denmark | Germany | -4.59 | <0.001 |
| Harm | Denmark | Greece | -4.78 | <0.001 |
| Harm | Denmark | Himalayas | -4.72 | <0.001 |
| Harm | Denmark | Hungary | -6.21 | <0.001 |
| Harm | Denmark | Ireland | -4.14 | 0.025 |
| Harm | Denmark | Italy | -5.29 | <0.001 |
| Harm | Denmark | Japan | -5.66 | <0.001 |
| Harm | Denmark | Philippines | -4.59 | <0.001 |
| Harm | Denmark | Romania | -5.05 | <0.001 |
| Harm | Denmark | Australia | -5.66 | <0.001 |
| Harm | Denmark | Canada | -7.88 | <0.001 |
| Harm | Germany | Norway | 4.62 | <0.001 |
| Harm | Greece | Norway | 4.48 | <0.001 |
| Harm | Himalayas | Norway | 4.60 | <0.001 |
| Harm | Hungary | Norway | 6.05 | <0.001 |
| Harm | India | Norway | 4.01 | 0.042 |
| Harm | Ireland | Norway | 4.19 | 0.022 |
| Harm | Italy | Norway | 5.14 | <0.001 |
| Harm | Japan | Norway | 5.38 | <0.001 |
| Harm | Norway | Philippines | -4.65 | <0.001 |
| Harm | Norway | Romania | -4.83 | <0.001 |
| Harm | Norway | Australia | -5.81 | <0.001 |
| Harm | Norway | Canada | -7.71 | <0.001 |
| Harm | Russia | Canada | -4.10 | 0.037 |
| Harm | Scotland | Canada | -4.27 | 0.025 |
| Harm | Canada | Brazil | 4.49 | 0.025 |
| Cheating | China | Spain | -4.48 | <0.001 |
| Cheating | Denmark | Spain | -4.63 | <0.001 |
| Cheating | England | Spain | -5.58 | <0.001 |
| Cheating | England | Turkey | -4.62 | <0.001 |
| Cheating | England | Canada | -4.38 | 0.011 |
| Cheating | Russia | Spain | -4.56 | <0.001 |
| Betrayal | England | Iceland | -4.41 | 0.011 |
| Authority | China | Denmark | 8.00 | <0.001 |
| Authority | China | England | 5.57 | <0.001 |
| Authority | China | France | 4.33 | 0.012 |
| Authority | China | Germany | 4.77 | <0.001 |
| Authority | China | Greece | 4.93 | <0.001 |
| Authority | China | Ireland | 4.91 | <0.001 |
| Authority | China | Italy | 5.36 | <0.001 |
| Authority | China | Norway | 7.13 | <0.001 |
| Authority | China | Russia | 4.83 | <0.001 |
| Authority | China | Serbia | 4.12 | 0.039 |
| Authority | China | Wales | 4.11 | 0.039 |
| Authority | China | Canada | 4.23 | 0.024 |
| Authority | China | Brazil | 4.26 | 0.022 |
| Authority | Denmark | Spain | -4.16 | 0.024 |
| Subversion | Germany | Africa | -4.78 | <0.001 |
| Sanctity | China | England | 4.71 | <0.001 |
| Sanctity | China | Greece | 4.81 | <0.001 |
| Sanctity | China | Italy | 4.79 | <0.001 |
| Sanctity | China | Norway | 5.28 | <0.001 |

Figure 8: Statistically significant differences (p < 0.05 after Bonferroni correction) in cultures by expression of Moral Foundations in folk tales.