# OpenReview forum: "Cross-Cultural Analysis of Human Values, Morals, and Biases in Folk Tales"
_EMNLP/2023/Conference — EMNLP 2023 Main_

### Official Review · Reviewer_Vntz · 2023-07-30

**Soundness:** 4

**Excitement:**

4: Strong: This paper deepens the understanding of some phenomenon or lowers the barriers to an existing research direction.

**Paper Topic And Main Contributions:**

This paper presents a cross-cultural study of value, morals, and gender biases in folk stories across a big number of countries. The authors perform analyses using English lexicon datasets for value/moral analyses and tools for dependency parsing and coreference resolution for gender bias analyses. To carry out the study, a dataset of English-translated folk tales from multiple sources was collected.

**Reasons To Accept:**

The paper is well written and the analyses performed by the authors are extensive and insightful. The work is the first of its kind to perform such a cross-cultural study across a wide range of countries, and the dataset created will be a valuable resource for future studies. The results also revealed multiple interesting insights, in particular related to what moral foundations are most reflected within the tales of each culture, while the gender bias analyses were less surprising.

**Reasons To Reject:**

The following are few concerns that the authors may be able to address:


Africa is considered as one culture, whereas the rest of the cultures considered are based on specific countries. The resource used for those tales states “West African Folk-Tales”, but it is not clear what countries are included in it, which could be problematic since, for example, Morocco and Ghana are completely different cultures.


Despite the paper mentioning that most previous work is focused on Anglo-American and European cultures, European countries are over-represented in the dataset (19 countries), compared to only 1 country (Brazil) from Latin America, 1 combined African culture, and no representation of Middle Eastern countries. I believe more representation of cultures from those areas would have allowed for a stronger cross-cultural study, although it is understandable that data may not be available. However, a suggestion would be the inclusion of tales from such cultures even if the number of tales available is less than 10.


**Reproducibility:**

4: Could mostly reproduce the results, but there may be some variation because of sample variance or minor variations in their interpretation of the protocol or method.

**Reviewer Confidence:**

4: Quite sure. I tried to check the important points carefully. It's unlikely, though conceivable, that I missed something that should affect my ratings.

---

> ### Author Rebuttal · Authors · 2023-08-28
>
> Thank you for your positive review of our work. We are glad you found our analyses to be extensive and the findings insightful.
>
> To address your comments:
>
> Cultures: We agree that Africa and the Middle East (and South America, Southeast Asia, and many more) are underrepresented in our dataset. Unfortunately, we are limited by our source: Project Gutenberg has very few books of tales from these regions. We will consider adding another book, South African Folk Tales, to the dataset.
>
> According to the introduction of West African Folk Tales, the stories are folklore of the “Gold Coast peoples”, which is modern-day Ghana (when this book was written, this area consisted of several territories rather than a single country).
>
> Including cultures with <10 tales: We actually considered this when performing our analyses but decided that the analyses would not be representative with such a small sample size. One possibility could be to include two versions of the dataset, one with cultures with >=10 tales, and another with all tales, and we will consider this for the camera-ready version.

---

### Official Review · Reviewer_irgT · 2023-08-01

**Soundness:** 4

**Excitement:**

3: Ambivalent: It has merits (e.g., it reports state-of-the-art results, the idea is nice), but there are key weaknesses (e.g., it describes incremental work), and it can significantly benefit from another round of revision. However, I won't object to accepting it if my co-reviewers champion it.

**Missing References:**

N/A

**Paper Topic And Main Contributions:**

This paper presents a dataset of 1925 folk tales representing 27 cultures and provides an analysis of human values, morals, and gender biases in individual tales across cultures. The authors argue that no such corpus currently exists, since existing works mainly focus on European tales.
For human values, the analysis consists of a lexicon-based approach using the Values Lexicon, which contains keywords for 49 human values. Experimental results show that the most common values expressed across cultures are social values, but that different cultures have tendencies to different human values. Moreover, the authors compare the occurrence of such value-representing words between novels (sourced from Project Gutenberg) and the folk tales, finding that folk tales have a stronger representation of such words.
For morals, the authors also employ a lexicon-based approach, using the Moral Foundations Dictionary 2.0 which contains keywords for 10 moral foundations. Here, it can be seen that positive aspects of morality are more strongly represented than negative ones and that morality is expressed differently across cultures.
For gender biases, the authors first identify the genders of characters in the tales, showing that male characters are overrepresented across cultures (2.29 times as many male characters as female ones) and that characters for both genders are associated with traditional values (e.g., children and family for female, career, wealth for male).


**Questions For The Authors:**

Did you conduct qualitative analyses to assess whether the scores assigned using the lexicon-based approaches are in line with your own estimations?

**Reasons To Accept:**

* This paper provides an extensive and rigorous analysis of a novel corpus of 1900 folk tales across 27 cultures.
* The introduced corpus, as well as the reported results, address a gap in the literature and are as such likely interesting to researchers working in this area.
* The paper is well-written, and the results are overall well-presented. Statistical testing is used to provide assessments of statistical significance across comparisons.


**Reasons To Reject:**

* The methods used in this paper have little to no technical novelty, and all experiments are based on lexicon-based approaches to measure textual characteristics. While this shows to provide interesting results, these methods have their limitations, and contextualized representations of text might be useful to better estimate concepts such as human values, morals, and gender biases.
* Related to the point above, it would have been interesting to see how human annotations correlate with the lexicon-based approach. The concepts measured in this work are complex, and a small-scale experiment to assess how well the lexicon-based approach measures them (based on how humans would evaluate these) would have strengthened the paper.


**Reproducibility:**

4: Could mostly reproduce the results, but there may be some variation because of sample variance or minor variations in their interpretation of the protocol or method.

**Reviewer Confidence:**

3: Pretty sure, but there's a chance I missed something. Although I have a good feel for this area in general, I did not carefully check the paper's details, e.g., the math, experimental design, or novelty.

**Typos Grammar Style And Presentation Improvements:**

* Line 285: one “are” too much in “…are they are…”
* For Figures 2 and 4, it would be helpful to see (normalized) numbers for each box, instead of only relying on the color intensity.
* I recommend not to refer to Tables in the Appendix from the main manuscript (e.g., Table 10) if they are crucial to understanding the results described in the text. The Appendix should serve as a space for supplementary material that is not essential to understand the main findings presented in the manuscript.

---

> ### Author Rebuttal · Authors · 2023-08-28
>
> Thank you for the positive comments on our work! We are glad you found our analyses extensive and rigorous, and the results well presented.
>
> To address your comments:
>
> Methods: The novelty of our paper lies in the large cross-cultural dataset and the insights drawn from applications of the methods across cultures, rather than in the methods themselves.
>
> Human annotations: The lexicons we employ have been validated against human annotations in prior work. For example, the values lexicon was validated against human-labeled word intrusion and category matching tasks, with up to 0.72 accuracy [1], and Moral Foundations Dictionary has been used in a classifier and tested against hand-annotated Twitter posts with 0.66 F1 score [2]. We will include a discussion of these points in the camera ready.
>
> [1] Wilson et al. (2018). Building and Validating Hierarchical Lexicons with a Case Study on Personal Values
> [2] Hoover et al. (2020). Moral foundations twitter corpus: A collection of 35k tweets annotated for moral sentiment
>
> Presentation: Thank you for the presentation suggestions. With the extra page in the camera ready, we can move Table 10 into the main content.

---

### Official Review · Reviewer_bzwS · 2023-08-05

**Soundness:** 4

**Excitement:**

4: Strong: This paper deepens the understanding of some phenomenon or lowers the barriers to an existing research direction.

**Paper Topic And Main Contributions:**

The paper compiles a dataset of folk tales and conducts a cross-cultural analysis of human values, morals, and biases. The computationally-aided linguistic analysis of folk tales is novel. The dataset can facilitate more reseach into this important research topic.

**Reasons To Accept:**

1. The computational approach to understanding folk tales is novel, compared to the large body of qualitative and literature research.
2. The dataset is carefully collected and valuable for future research.
3. The findings are interesting.

**Reasons To Reject:**

1. There's a lack of disucssion regarding why Human Values, Morals, and Biases are discussed at the same time.
2. There's a lack of engagement with prior literature in computationally understanding bias or values in literature. For example, [1] is a paper on fairy tales and has some similar findings about gender bias and moral foundations: "female characters turn out more associated with care-, loyalty- and sanctity- related moral words, while male characters are more associated with fairness- and authority- related moral words." Engaging with such literature will further extend the paper's contribution to computationally understanding values and biases in literature.

[1] Zhou, Z., Sun, J., Pei, J., Peng, N., & Xiong, J. (2022). A Moral-and Event-Centric Inspection of Gender Bias in Fairy Tales at A Large Scale. arXiv preprint arXiv:2211.14358.

**Reproducibility:**

4: Could mostly reproduce the results, but there may be some variation because of sample variance or minor variations in their interpretation of the protocol or method.

**Reviewer Confidence:**

5: Positive that my evaluation is correct. I read the paper very carefully and I am very familiar with related work.

---

> ### Author Rebuttal · Authors · 2023-08-28
>
> Thank you for your positive review of our work. We are glad you found our findings interesting.
>
> To address your comments:
>
> We study human values, morals, and biases because they are closely intertwined, and they influence one another. In addition, in our work, we found that these three aspects can be analyzed using similar methods.
>
> Thank you for the Zhou et al. (2022) reference. Indeed, they found similar results, although on a significantly smaller set of seven cultures (as compared to the 27 diverse cultures included in our study). We also note that this paper has not been peer-reviewed, but we will include it in the camera-ready.

---

### Meta-Review · Area_Chair_pGT1 · 2023-09-24

**Recommendation:** 4

**Metareview:**

This paper compiles a large corpus of over 1,900 tales from 27 diverse cultures across six continents, and conducts language and statistical analyses to study how human values, morals, and gender biases are expressed in folk tales across cultures. Three reviewers reviewed the submission, highlighted the strengths of the paper in terms of novelty (R1, R3), data contributions (R1, R2), presentation and readability (R2, R3), and findings (R1, R2, R3). During the rebuttal phase, the authors also responded that they would address the reviewers’ concerns through engagement with prior literature and additional analyses.

---

### Decision · Program_Chairs · 2023-10-07

**Decision:**

Accept-Main

**Comment:**

This paper compiles a large corpus of over 1,900 tales from 27 diverse cultures across six continents, and conducts language and statistical analyses to study how human values, morals, and gender biases are expressed in folk tales across cultures. Three reviewers reviewed the submission, highlighted the strengths of the paper in terms of novelty (R1, R3), data contributions (R1, R2), presentation and readability (R2, R3), and findings (R1, R2, R3). During the rebuttal phase, the authors also responded that they would address the reviewers’ concerns through engagement with prior literature and additional analyses.